# GI-GCN: Global Interacted Graph Convolutional Networks via Dominant Sets for Graph Classification

Lu Bai [1]   Xinya Qin [1]   Lixin Cui [2]   Ming Li [3]   Hangyuan Du [4]   Ziyu Lyu [5]   Xin Jin [2]

## Abstract

Graph Convolutional Networks (GCNs) are defined based on aggregating the information of adjacent nodes, that are usually treated as equally important and may limit the representational power of existing GCNs. To address this shortcoming, we propose a novel Global Interacted Graph Convolutional Network (GI-GCN), that leverages the solution vectors maintained during the iterative updates of the Dominant Set to adaptively characterize the global importance distribution over all nodes. Specifically, at each convolution layer, this distribution is adopted to adaptively modulate the importance weights of node features before performing the local message passing. We show that this convolution strategy can effectively capture the highly correlated information between nonadjacent nodes through the Dominant Set algorithm, not only emphasizing the critical graph-level information but also enhancing the discriminative power of graph representations. Furthermore, we optimize the memory complexity of the framework, significantly reducing the memory overhead associated with the global interaction modeling. Experiments demonstrate the effectiveness of the proposed GI-GCN model.

[1]School of Artificial Intelligence, Beijing Normal University, Beijing, China (bailu@bnu.edu.cn; XinyaQin@mail.bnu.edu.cn). [2]School of Information, Central University of Finance and Economics, Beijing, China (cuilixin@cufe.edu.cn). [3]Zhejiang Institute of Optoelectronics, Zhejiang Key Laboratory of Intelligent Education Technology and Application, Zhejiang Normal University, Jinhua, China. [4]School of Computer and Information Technology, Shanxi University, Taiyuan, China. [5]School of Cyber Science and Technology, Sun Yat-Sen University, Shenzhen, China. Correspondence to: Hangyuan Du <duhangyuan@sxu.edu.cn>.

*Proceedings of the $43^{rd}$ International Conference on Machine Learning*, Seoul, South Korea. PMLR 306, 2026. Copyright 2026 by the author(s).

## 1. Introduction

In machine learning, graph-based representations are powerful tools for structured data analysis, and have been widely employed for various real-world applications, e.g., bioinformatics (Jing et al., 2021; Yan et al., 2023), social networks (McAuley & Leskovec, 2012; Min et al., 2021), and recommendation systems (Chen et al., 2022b; Wang et al., 2023). One challenge arising in graph data analysis is how to learn numeric characteristics for graph structures. To address this issue, Graph Neural Networks (GNNs) have been developed as a general framework for learning on graph-structured data, and have demonstrated the effective performance on various graph tasks, such as node classification, graph classification, and link prediction.

Among the various GNN architectures, Graph Convolutional Networks (GCNs) (Kipf & Welling, 2017) have emerged as one of the most representative models. Broadly speaking, GCNs generate expressive graph representations by aggregating node features according to the adjacency structure of the graph. From a spectral perspective, they can be interpreted as a first-order approximation of localized spectral filters on graphs. In practice, a GCN linearly transforms node features and then multiplies them by a normalized adjacency matrix with self-loops, so that each node is updated using itself and its one-hop neighbors. However, this equal-weighted local aggregation makes it difficult to capture the discriminative node importance from a global perspective, limiting the discriminative power of most existing GCNs.

To alleviate this limitation, previous studies have explored weighted or global aggregation mechanisms. For example, Graph Attention Networks (GATs) (Velickovic et al., 2018; He et al., 2021; Kim & Oh, 2021; Brody et al., 2022; Zhao et al., 2025) assign attention scores to neighboring nodes and adaptively modulate their contributions during the message aggregation. However, since the attention computation of GATs is still restricted to local neighborhoods, such methods have limited capability in capturing interactions among distant nodes. This in turn constrains their effectiveness in modeling the global structural information for graph-level tasks. In addition, Transformer-based graph models (Dwivedi & Bresson, 2021; Ying et al., 2021; Rampášek et al., 2022) explicitly model dependencies across the entire

graph, enabling more comprehensive global information exchange. Despite their strong representational capacity, these approaches typically rely on a substantial number of additional learnable parameters, resulting in significantly increased computational and memory costs.

More unfortunately, the above problems may seriously influence the effectiveness of most existing GCNs, especially for molecular graph classification. This is due to the fact that the required graph labels are often determined by a few discriminative functional groups or long-range interactions among distant atoms. Such tasks require both local structural inductive biases and effective global interaction modeling. This observation motivates a lightweight global interaction mechanism that can either identify the graph-level critical nodes or preserve the local structural information.

The aim of this paper is to address the aforementioned drawbacks by proposing a novel GNN associated with the Dominant Set analysis, instead of the classical attention and Transformer strategies. Specifically, the Dominant Set method is an optimization framework based on iterative dynamics that can identify the highly cohesive and externally dissimilar dominant node subsets within a given graph structure. We observe that this property can well align with our goal of modeling interactions among globally important nodes, and has been successfully employed for some of existing graph clustering (Pavan & Pelillo, 2007) and graph pooling (Ali et al., 2026). In this work, we further interpret the solution vector maintained during the iterative updates of Dominant Set as a natural encoder of dominance relationships between pairwise nodes, generating an importance distribution over all nodes at the global graph level. Since this importance distribution gradually converges toward a subset of dominant nodes, it can reflect the dominant global interaction between nodes, thereby providing an elegant way to effectively learn graph representations associated with global guidance signals of this distribution.

Inspired by the above observation, we propose a Global Interacted Graph Convolutional Network (GI-GCN). The key innovation of the proposed GI-GCN is to tightly integrate the iterative dynamics of Dominant Set into the graph convolution procedure, thereby constructing novel GNNs associated with the dominant global interaction capabilities. This design enables the node importance to be dynamically propagated across the entire graph and effectively guides the subsequent feature aggregation, providing a more discriminative global modeling paradigm for graph-level representation learning. Moreover, in contrast to the aforementioned methods that rely on parameterized attention mechanisms (i.e., the GNNs based on attention and transformer strategies), the Dominant Set does not introduce additional learnable attention parameters, naturally capturing the correlations between nodes that are not adjacent in

the graph topology but highly related in the feature space. Overall, the main contributions of this work are threefold.

**First**, we define a novel graph convolutional operation for the proposed GI-GCN that integrates the iterative dynamics of the Dominant Set into the convolution process in a layer-wise manner. Since the Dominant Set dynamically updates the graph-level node importance distribution, node features can be adaptively amplified or suppressed according to the dominant global interactions among nodes. As a result, unlike most existing GNNs, the proposed GI-GCN can effectively capture interactions between nodes that are not adjacent in the graph topology but highly correlated in the feature space. Moreover, with the Dominant Set dynamics progressively converging, the GI-GCN further emphasizes globally dominant nodes, thus resulting in more discriminative graph representations.

**Second**, we introduce a memory-efficient equivalent reparameterization that realizes the Dominant Set update in the feature dimension. This strategy significantly reduces the additional storage overhead from $O(n^2)$, that scales quadratically with the number of nodes, to $O(f^2)$, that depends only on the square of the feature dimension.

**Finally**, we conduct extensive experiments on multiple benchmark graph classification datasets, demonstrating that the proposed GI-GCN achieves competitive performance.

The remainder of the paper is organized as follows. Section 2 reviews related works on the attention mechanism and the Dominant Set method. Section 3 presents details of the proposed GI-GCN model. Section 4 reports experimental setups and results, and Section 5 concludes this work.

## 2. Related Works

In this section, we briefly review some related works and introduce the concept of the Dominant Set method.

### 2.1. Attention Mechanisms on Graphs

To enhance the information aggregation of GNNs, various attention-based weighted aggregation mechanisms have been proposed. For instance, GATs (Velickovic et al., 2018) introduce the learnable attention functions to adaptively reweight neighboring nodes, alleviating the representational limitations induced by the uniform aggregation. Subsequent studies have further extended the attention modeling with more expressive self-supervised schemes (e.g., the SuperGAT (Kim & Oh, 2021) and dynamic variants (e.g., GATv2 (Brody et al., 2022)) to improve the robustness against noisy graph structures. These works demonstrate the effectiveness of attention mechanisms in distinguishing the relative contributions of neighboring nodes and enhancing the node representations. However, most of these

attention-based methods tend to be restricted within the local neighborhoods, limiting their ability to capture global interactions among topologically distant nodes.

To explicitly model the global dependencies in graphs, the Transformer (Vaswani et al., 2017) architecture has been introduced for graph representation learning (Dwivedi & Bresson, 2021). For instance, Graphormer (Ying et al., 2021) incorporates structural encodings into the Transformer framework to inject the graph structural information, achieving strong performance on graph-level tasks. The Structure-Aware Transformer (SAT) (Chen et al., 2022a) further enhances the structural awareness by explicitly integrating subgraph- or structure-level features into the attention mechanism. In addition, GraphGPS (Rampášek et al., 2022) decouples the local message passing from global self-attention modules, balancing the local structural modeling with scalable global information exchanges. Other representative graph Transformer variants also include the Graphormer-GD (Zhang et al., 2023), GRIT (Ma et al., 2023), GraphViT (He et al., 2023), etc.

Despite the strong capability in modeling global dependencies, most graph Transformer-based methods rely on parameterized pairwise attention mechanisms, resulting in considerable parameter overhead and quadratic memory complexity with respect to the number of nodes. By contrast, the proposed GI-GCN follows a fundamentally different global modeling paradigm. Instead of introducing learnable global attentions, the GI-GCN derives a global node-importance distribution through Dominant Set dynamics, an optimization-driven process that requires no additional attention parameters. This design enables efficient global interaction modeling while retaining the local aggregation advantages of graph convolutions.

### 2.2. The Dominant Set Analysis

The Dominant Set method is an optimization-based graph analysis framework grounded in continuous optimizations and iterative dynamics, and was originally proposed to identify node subsets with either the high internal coherence or the low external similarity in weighted graphs. Specifically, it formulates the node selection as a quadratic programming problem over the probability simplex and solves it through iterative updates of a solution vector, that gradually converges to a dominant subset. Due to its rigorous optimization foundation, the resulting solution naturally provides a continuous node-importance distribution over the entire graph. Detailed descriptions of the Dominant Set algorithm are provided in Appendix A.

From an application perspective, the Dominant Set method has been extensively adopted for graph clustering (Pavan & Pelillo, 2007), image segmentation (Mequanint et al., 2019), and graph pooling (Ali et al., 2026), that usually require to identify structurally compact or semantically coherent node subsets. However, for most of these studies, the Dominant Set is primarily employed as an offline tool for the subset selection or structural compression. By contrast, this work exploits its dynamic information embedded during the iterative update process. Specifically, we employ its solution vector maintained during the iterations as a global node-importance distribution, that progressively concentrates on dominant nodes. Such dynamics can naturally align with our objective of learning graph-level global interaction priors, providing an effective guidance signal for graph representation learning. Moreover, the Dominant Set introduces no additional learnable attention parameters. Instead, the node importance is entirely determined by the feature similarity as well as the iterative dynamics, yielding a more interpretable and parameter-efficient global modeling mechanism.

## 3. The Proposed GI-GCN via Dominant Sets

In this section, we give the definition of the proposed GI-GCN model associated with the Dominant Set analysis.

### 3.1. The Overall Framework of the Proposed GI-GCN

Figure 1(a) illustrates the overall architecture of the proposed GI-GCN model, that consists of multiple stacked GI-GCN layers followed by a graph-level average readout function. Specifically, the GI-GCN adopts a layer-wise global interaction modeling strategy, where the iterative dynamics of the Dominant Set are computed at each layer to dynamically maintain and update the node importance distributions. These distributions serve as global guidance signals that modulate the subsequent feature propagation and aggregation, enabling the model to progressively focus on the dominant nodes with discriminative power. Under this scenario, the GI-GCN can explicitly either enhance the global discriminative capability of graph representations or preserve the ability of capturing local structural information.

Figure 1(b) further illustrates the computation pipeline of a single GI-GCN layer, that mainly consists of three steps as follows. **Step 1: The Node correlation modeling.** Based on the node representations at the current layer, a node correlation matrix is constructed to capture the correlations between arbitrary pairs of nodes. This process is independent of the explicit graph topology, allowing the model to encode the potential non-local interactions among nodes. **Step 2: The Node importance learning via Dominant Set dynamics.** Given the correlation matrix, Dominant Set replicator dynamics are employed to iteratively learn a node importance distribution defined over the entire graph structure. This distribution characterizes the relative importance and dominance relationships among nodes, and progressively concentrates on a subset of dominant nodes during the iterative process. **Step 3: The Importance modula-**

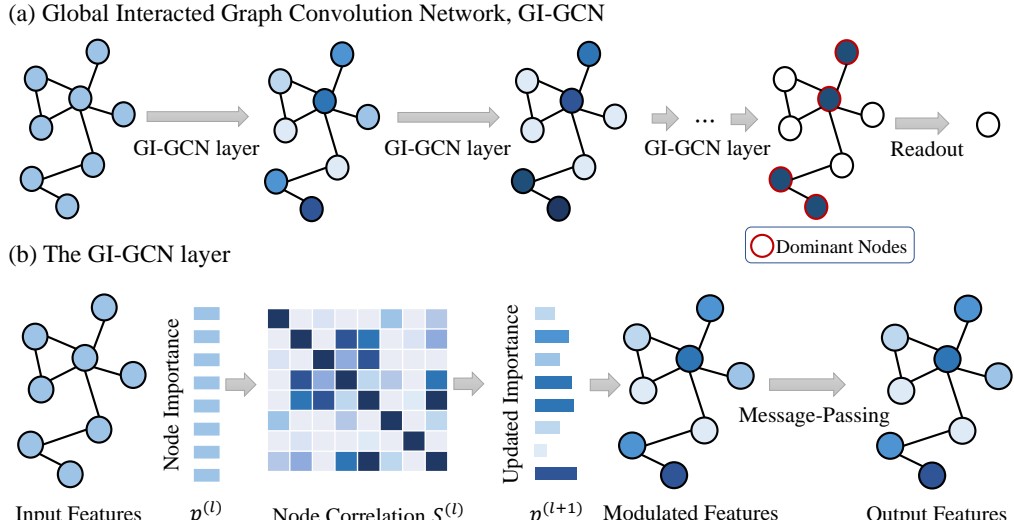

*Figure 1.* The framework of the proposed GI-GCN.

**tion and graph convolution.** The learned node importance distribution is then used to adaptively modulate the node features, followed by a standard graph convolution operation. In this manner, the global interaction information is explicitly injected into the local message passing. Through the coordination of the three steps, the GI-GCN achieves an effective integration of the global node importance modeling and the local structural aggregation. The following subsections provide detailed definitions of each step.

### 3.2. The Node Correlation Modeling

Given a sample graph $G(\mathcal{V}, \mathcal{E}, A, X)$, where $\mathcal{V}$ ($|\mathcal{V}| = n$) denotes the node set, $\mathcal{E}$ denotes the edge set, $A \in \mathbb{R}^{n \times n}$ is the adjacency matrix, and $X \in \mathbb{R}^{n \times f}$ is the node feature matrix. For attributed graphs, the node features are initialized using one-hot encodings of node labels. For unattributed graphs, we use one-hot encodings of node degrees instead. The goal of graph classification is to learn a mapping function that produces the discriminative graph-level representations.

At the $l$-th layer, the node representations are denoted as $H^{(l)} = [h_1^{(l)}; \ldots; h_n^{(l)}]$, where $h_i^{(l)} \in \mathbb{R}^{1 \times f}$ represents the feature vector of node $i$ ($i = 1, \ldots, n$). For the initial layer, the node representations are directly initialized from the input node feature matrix $X$ (i.e., $H^{(0)} = X$). To capture the potential correlations among nodes in the feature space, we construct a feature-induced node correlation matrix. Each node feature vector is first centered and normalized, i.e.,

$$\hat{h}_i^{(l)} = \frac{h_i^{(l)} - \mu_i^{(l)}}{\sigma_i^{(l)}}, \tag{1}$$

where $\mu_i^{(l)}$ and $\sigma_i^{(l)}$ denote the mean and standard deviation computed over the feature dimensions of node $i$ at layer

$l$. We then compute the Pearson correlation coefficient between node $i$ and node $j$ as

$$C_{ij}^{(l)} = \hat{h}_i^{(l)} \big(\hat{h}_j^{(l)}\big)^{\top}, \quad C_{ij}^{(l)} \in [-1, 1]. \tag{2}$$

The Pearson correlation effectively captures the linear relationships between the node features and is robust to feature scale differences. To satisfy the non-negativity requirement of the Dominant Set optimization, the correlations are transformed into the $[0, 1]$ range using a quadratic mapping as

$$S_{ij}^{(l)} = C_{ij}^{(l)^2}, \tag{3}$$

resulting in the node correlation matrix $S^{(l)} \in [0, 1]^{n \times n} \subset \mathbb{R}^{n \times n}$. Notably, this correlation matrix is entirely feature-driven and independent of the explicit graph topology, allowing the model to capture potential relationships between nodes that are not adjacent in the graph structure space but highly correlated in the feature space. More specifically, it will serve as the necessary input for the subsequent Dominant Set-based global node importance learning, thereby introducing a global interaction prior into the graph convolution process. Note that, this does not imply that the GI-GCN ignores the graph topology. The correlation matrix is solely introduced to estimate a global node importance distribution, that can modulate node representations before graph convolution operations (see Section 3.4). The structural dependencies among neighboring nodes are still explicitly modeled through the adjacency matrix during the message passing. Consequently, the GI-GCN jointly captures global feature correlations and local topological interactions, rather than relying solely on feature similarities.

### 3.3. The Node Importance Learning via Dominant Sets

The Dominant Set algorithm is a similarity-based continuous optimization framework originally proposed for identifying highly cohesive node subsets, e.g., maximal cliques, in graphs (Pavan & Pelillo, 2007). It follows the principle of high intra-cluster similarity and low inter-cluster similarity, allowing nodes to be adaptively grouped or separated.

Given the node set $\mathcal{V} = \{1, \ldots, n\}$ and the similarity matrix $S \in \mathbb{R}^{n \times n}$, where $S_{ij}$ denotes the similarity between nodes $i$ and $j$, and $S$ is typically assumed to be non-negative and symmetric, the Dominant Set problem can be formulated as the following continuous optimization

$$\max_{p} \; p^\top S p \quad \text{s.t.} \quad p \in \Delta, \tag{4}$$

where $\Delta = \{p \in \mathbb{R}^n \mid p_i \geq 0, \sum_i p_i = 1\}$ denotes the standard simplex. $\Delta$ can ensure that the solution vector $p$ can be naturally interpreted as a probability distribution over all nodes, thereby characterizing their relative importance. Since this objective takes the form of a constrained quadratic program, it is commonly solved via the replicator dynamics (Weibull & Press, 1995). This iterative optimization scheme preserves the feasibility on the simplex while monotonically increases the objective value, providing a stable and efficient mechanism for Dominant Set learning. Moreover, the discrete update rule of the solution vector $p$ associated with different $t$ iterations is given by

$$p_i^{(t+1)} = p_i^{(t)} \frac{(S p^{(t)})_i}{p^{(t)\top} S p^{(t)}}, \quad i = 1, \ldots, n. \tag{5}$$

Under mild conditions, this iteration converges into a local maximizer of the quadratic program while preserves feasibility on the simplex. The resulting solution vector $p$ highlights a subset of representative nodes, reflecting their relative importance and dominance relationships.

In this work, we interpret the solution vector $p$ computed during the iterative updates at each convolutional layer as a global node importance distribution. This distribution progressively concentrates on a subset of dominant nodes, thereby providing a stable and effective global guidance signal for graph-level representation learning. By leveraging this signal, the model can adaptively emphasize key nodes during the feature aggregation, enhancing the discriminative power of graph representations. Notably, this mechanism is fundamentally different from conventional mask-based strategies. Mask strategies are typically discrete structural operations, such as directly removing nodes or edges, or restricting the information propagation through binary masks. By contrast, the Dominant Set of the proposed GI-GCN produces a continuous probability distribution defined on the simplex, which is derived from an optimization process jointly driven by global structural relationships and node

features. This distribution reflects the relative importance of all nodes across the entire graph structure.

Specifically, we initialize the node importance vector at the first convolutional layer as a uniform distribution, i.e., $p^{(0)} = \frac{1}{n}\mathbf{1}$, indicating that all nodes are equally important at the beginning. Subsequently, the importance vector is updated jointly with node representations at each convolutional layer. Given the node importance vector $p^{(l)}$ at layer $l$ and the corresponding node correlation matrix $S^{(l)}$ computed in Section 3.2, the next-layer importance vector is updated as

$$p_i^{(l+1)} = p_i^{(l)} \frac{(S^{(l)} p^{(l)})_i}{p^{(l)\top} S^{(l)} p^{(l)}}, \quad i = 1, \ldots, n. \tag{6}$$

This layer-wise update mechanism ensures that the global node importance information is progressively propagated and amplified across the graph, allowing the model to adaptively highlight the dominant nodes and enhance the discriminative capacity of the graph-level representations.

### 3.4. The Importance Modulation & Graph Convolution

After obtaining the node importance distribution $p^{(l)} \in \mathbb{R}^n$ at the $l$-th layer, where $p_i^{(l)}$ denotes the global importance score of node $i$, the proposed GI-GCN utilizes $p^{(l)}$ to adaptively modulate node features. Specifically, the node representations are reweighted in an element-wise manner as

$$\tilde{h}_i^{(l)} = p_i^{(l)} \cdot h_i^{(l)}, \quad i = 1, \ldots, n, \tag{7}$$

where $\tilde{h}_i^{(l)}$ denotes the modulated node representation. This modulation explicitly amplifies the features of nodes with higher global importance while suppressing less important ones, thereby injecting the global interaction information before the local message passing and enhancing the discriminative capacity of the resulting graph representations.

Subsequently, the GI-GCN performs a standard graph convolution operation on the modulated node features as

$$H^{(l+1)} = \sigma\left(\tilde{A} \tilde{H}^{(l)} W^{(l)}\right), \tag{8}$$

where $\tilde{H}^{(l)} = [\tilde{h}_1^{(l)}; \ldots; \tilde{h}_n^{(l)}]$ denotes the matrix of modulated node features, $\tilde{A} \in \mathbb{R}^{n \times n}$ is the normalized adjacency matrix with added self-loops, $W^{(l)} \in \mathbb{R}^{f \times f'}$ is the learnable weight matrix, and $\sigma(\cdot)$ is the nonlinear activation function.

It is important to emphasize that the proposed GI-GCN does not modify the original graph structure or explicitly remove any edge. The associated importance modulation only re-weights node features, while the message passing is still performed on the original graph topology through the adjacency matrix. Therefore, the GI-GCN fully preserves the graph structure and ensures that all edges consistently

participate in the feature propagation. In this way, the global node importance learned via Dominant Set dynamics effectively guides local feature aggregation, allowing non-local correlation information to be continuously injected into the graph convolution process without any structural pruning.

## 3.5. The Memory-Efficient Reformulation

The primary space complexity of the GI-GCN arises from the explicit construction and storage of the node-wise correlation matrix $S^{(l)} \in \mathbb{R}^{n \times n}$ at layer $l$. In the feature-induced correlation modeling, node features are first centered and normalized, and pairwise correlations are then mapped into the range $[0, 1]$ to satisfy the non-negativity requirement of the Dominant Set formulation. In this setting, the key term of the replicator dynamics can be expressed as

$$(S^{(l)}p^{(l)})_i = \sum_j \left(\hat{h}_i^{(l)}(\hat{h}_j^{(l)})^\top\right)^2 p_j^{(l)}, \qquad (9)$$

where $\hat{h}_i^{(l)} \in \mathbb{R}^{1 \times f}$ denotes the normalized feature vector of node $i$, and $p^{(l)}$ is the node importance distribution at layer $l$. To avoid explicitly storing the matrix $S^{(l)}$, the above expression can be exactly reformulated as

$$\begin{aligned}(S^{(l)}p^{(l)})_i &= \sum_j \hat{h}_i^{(l)}(\hat{h}_j^{(l)})^\top \hat{h}_j^{(l)}(\hat{h}_i^{(l)})^\top p_j^{(l)} \\ &= \hat{h}_i^{(l)}\left(\sum_j p_j^{(l)}(\hat{h}_j^{(l)})^\top \hat{h}_j^{(l)}\right)(\hat{h}_i^{(l)})^\top.\end{aligned} \qquad (10)$$

By defining

$$M^{(l)} = \sum_j p_j^{(l)}(\hat{h}_j^{(l)})^\top \hat{h}_j^{(l)} \in \mathbb{R}^{f \times f}, \qquad (11)$$

the update for each node reduces to a quadratic form in the feature space, eliminating the need to construct the node-level correlation matrix. This reformulation is fully equivalent to the original Dominant Set dynamics while reducing the space complexity from $\mathcal{O}(n^2)$ to $\mathcal{O}(f^2)$. Since the feature dimension $f$ is typically much smaller than the number of nodes $n$ in graph learning tasks, this design substantially improves the memory efficiency and scalability of GI-GCN.

## 3.6. The Properties of the Proposed GI-GCN Model

The GI-GCN exhibits notable advantages in the adaptive node importance modeling, global interaction-aware graph convolution, and efficient scalable implementation. These properties are analyzed from the following aspects.

**The Adaptive Node Importance Learning.** The proposed GI-GCN incorporates the replicator dynamics of the Dominant Set into each convolutional layer to adaptively learn a node importance distribution over the entire graph. This dynamic process evolves on the probability simplex, and

converges to a local optimum of the associated quadratic optimization problem under the mild conditions, causing the importance distribution to progressively concentrate on the dominant nodes. Rather than producing a hard selection, it yields a continuous soft distribution that reflects the relative importance of nodes across the graph. Such nodes typically exhibit stronger coherence and representativeness in the feature space, characterizing the global structure and semantics of the graph and naturally guiding the model to focus on the most discriminative nodes for graph-level prediction.

**The Global Interacted Graph Convolution.** Unlike the attention-based methods that compute weights only within local neighborhoods, the node importance distribution of the proposed GI-GCN is derived from a global correlation matrix, enabling direct modeling of arbitrary node-pair relationships without being constrained by graph adjacency. Consequently, each convolutional layer inherently supports cross-neighborhood interactions and long-range dependency modeling, alleviating the expressiveness limitations of purely local message-passing schemes. At the same time, the GI-GCN performs feature aggregation along the graph topology through standard graph convolution, effectively preserving the structural information of the graph. By injecting the global node importance into the local message passing, the convolution integrates global semantic cues with local structural information, progressively enhancing the representations of key nodes and their substructures.

**The Efficient and Scalable Implementation.** In GI-GCN, the node importance is inferred from feature-induced similarity and intrinsic iterative dynamics, avoiding additional learnable attention weights and improving parameter efficiency and training stability compared with Transformer-based graph models. Furthermore, by reformulating the key computations of the Dominant Set updates from the node-pair space to the feature space, GI-GCN reduces the space complexity from $\mathcal{O}(n^2)$ to $\mathcal{O}(f^2)$, substantially improving memory efficiency and scalability. This makes global interaction modeling feasible for large-scale graphs. More importantly, this design is different from virtual-node communication, spectral global convolution, spatial propagation, and linear-attention approximation, because the global interaction is derived from an optimization process rather than architectural augmentation or attention approximation.

## 4. Experiments

In this section, we conduct extensive experiments to evaluate the effectiveness of the proposed GI-GCN on benchmark graph classification tasks. Specifically, we compare its classification performance with representative graph kernel methods and GNN baselines, analyze its computational efficiency and memory consumption, and provide qualitative visualizations to illustrate the learned node importance

distributions. These experiments aim to demonstrate both the empirical effectiveness and the practical scalability of GI-GCN. Additional experimental results and analyses, including detailed ablation studies, sensitivity evaluations, and large-scale benchmark comparisons, are provided in Appendices B and C.

## 4.1. The Experimental Setup

To evaluate the effectiveness of the proposed GI-GCN, we conduct experiments on multiple widely used benchmark graph classification datasets covering two major application domains,i.e., bioinformatics (Bio) and social networks (SN). Table 1 summarizes the maximum number of nodes (Max #Nodes), average number of nodes (Mean #Nodes), total number of graphs (#Graphs), number of classes (#Classes), and corresponding application domain for each dataset.

All experiments are implemented using PyTorch and PyTorch Geometric, and conducted on a single NVIDIA GeForce RTX 3090 GPU with 24GB of VRAM. For the proposed GI-GCN, the node importance distribution is initialized as a uniform distribution at the first layer. The number of graph convolutional layers is selected from 2 to 4. The mean readout function is adopted to obtain graph-level representations, and the node embedding dimension is fixed at 32. We employ the ReLU activation function and optimize the models using the Adam optimizer. The learning rate is selected from $\{0.001, 0.0005, 0.0001\}$, while the weight decay is chosen from $\{10^{-5}, 10^{-4}, 10^{-3}\}$. The dropout rate varies from 0 to 0.5, depending on the dataset. The number of training epochs ranges from 100 to 1000 according to the dataset scale and convergence behavior. To ensure statistical robustness, we perform 10 repetitions of 10-fold cross-validation and report the average classification accuracy together with the standard error. The implementation code is publicly available on our GitHub repository.[1]

## 4.2. The Graph Classification Performance

We compare the proposed GI-GCN with a broad range of representative graph kernel methods and GNN models to comprehensively evaluate its performance. The graph kernel methods include the RWGK (Gartner et al., 2003), SPGK (Borgwardt & Kriegel, 2005), WLSK (Shervashidze et al., 2011), JTQK (Bai et al., 2014) (with $q = 2$), EDBMK (Xu et al., 2021), QBMK (Bai et al., 2024), GK-A, GK-X, GK-C (Qin et al., 2025), and the AEGK (Bai et al., 2025), which mainly rely on explicit or implicit structural similarity modeling and have demonstrated strong performance in graph classification tasks. The GNN baselines include the DGCNN (Zhang et al., 2019), DiffPool (Ying et al., 2018), ECC (Simonovsky & Komodakis, 2017),

GIN (Xu et al., 2019), GraphSAGE (Hamilton et al., 2017), GraphiT (Mialon et al., 2021), GWRNN (Nikolentzos & Vazirgiannis, 2023), Co-GNN (Finkelshtein et al., 2024), GKNN-WL, GKNN-GL (Cosmo et al., 2025), and the DSMVPool (Ali et al., 2026). For the DGCNN, DiffPool, ECC, GIN, and GraphSAGE, we adopt the results reported in (Errica et al., 2020) or evaluate them under the same experimental settings for datasets not originally reported. For the remaining methods, we report the results from their original publications.

The classification performance comparisons are reported in Table 2. Overall, the proposed GI-GCN achieves competitive or superior classification accuracy on the majority of benchmark datasets. In particular, it consistently outperforms the competing methods on the PROTEINS, DD, IMDB-B, and IMDB-M datasets, demonstrating its effectiveness and robustness across different application domains and graph structural complexities. These performance improvements can be attributed to the following aspects.

First, compared with graph kernel methods, our GI-GCN benefits from an end-to-end feature learning paradigm that jointly optimizes the node representations and the graph-level classification objectives. Rather than relying on hand-crafted structural statistics or predefined substructure similarity measures, our GI-GCN learns task-driven node embeddings that more effectively capture the complex interplay between the node attributes and the global graph structure, thereby yielding more discriminative graph representations.

Second, in comparison with the existing GNN-based approaches, the key advantage of the GI-GCN lies in its explicit modeling of the global node importance. By leveraging the Dominant Set dynamics, the GI-GCN maintains a globally defined node importance distribution across the entire graph, highlighting the nodes and substructures that contribute most significantly to the graph-level discrimination task. This mechanism enables the model to progressively focus on the representative components of the graph across multiple convolutional layers, instead of treating all nodes uniformly or relying solely on the local neighborhood-level attention mechanisms.

Third, the learned global importance information is adaptively injected into the local message passing process. This design effectively combines global interaction modeling with local structural aggregation, alleviating the limitations of purely local propagation schemes in capturing long-range dependencies. As a result, GI-GCN achieves robust and consistent classification improvements across diverse datasets, particularly in scenarios where accurate graph classification depends on identifying critical nodes or substructures.

---

[1] https://github.com/Xiaoqin0421/GI-GCN

*Table 1.* Statistics of benchmark graph classification datasets.

|  | MUTAG | PTC_MR | PROTEINS | DD | IMDB-B | IMDB-M |
|---|---|---|---|---|---|---|
| Max #Nodes | 28 | 64 | 620 | 5748 | 136 | 89 |
| Mean #Nodes | 17.93 | 14.29 | 39.06 | 284.30 | 19.77 | 13.00 |
| Max #Edges | 33 | 71 | 1049 | 14267 | 1249 | 1467 |
| Mean #Edges | 19.79 | 14.69 | 78.82 | 715.66 | 96.53 | 65.94 |
| #Graphs | 188 | 344 | 1113 | 1178 | 1000 | 1500 |
| #Classes | 2 | 2 | 2 | 2 | 2 | 3 |
| Domain | Bio | Bio | Bio | Bio | SN | SN |

*Table 2.* Graph classification accuracy (% $\pm$ standard error) on benchmark datasets.

| Category | Method | MUTAG | PTC_MR | PROTEINS | DD | IMDB-B | IMDB-M |
|---|---|---|---|---|---|---|---|
| **Ours** | **GI-GCN** | 87.99±1.44 | 59.16±1.76 | **79.97±1.37** | **88.24±1.74** | **80.58±2.19** | **55.91±0.50** |
| Graph Kernels | RWGK (2003) | 80.77±0.72 | 55.91±0.37 | 74.20±0.40 | 71.70±0.47 | 67.94±0.77 | 46.72±0.30 |
|  | SPGK (2005) | 83.38±0.31 | 56.55±0.53 | 75.10±0.50 | 78.45±0.26 | 71.26±1.04 | 51.33±0.57 |
|  | WLSK (2011) | 82.88±0.57 | 56.05±0.51 | 73.52±0.43 | 79.78±0.36 | 71.88±0.77 | 49.50±0.49 |
|  | JTQK (2014) | 85.50±0.55 | 57.39±0.46 | 72.86±0.41 | 79.49±0.32 | 72.45±0.81 | 50.33±0.49 |
|  | EDBMK (2021) | 86.35 | 56.75 | – | 78.19 | – | – |
|  | QBMK (2024) | 88.55±0.43 | 59.38±0.36 | – | 77.60±0.47 | – | – |
|  | GK-A (2025) | 88.72±0.39 | 58.29±0.71 | 75.18±0.31 | 78.65±0.27 | 73.19±0.23 | 50.20±0.20 |
|  | GK-X (2025) | 88.50±0.85 | **61.24±0.50** | 75.42±0.41 | 79.16±0.41 | 74.1±0.37 | 50.23±0.27 |
|  | GK-C (2025) | 87.39±0.69 | 60.71±0.28 | 74.95±0.31 | 80.03±0.26 | 73.37±0.33 | 50.29±0.22 |
|  | AEGK (2025) | **89.11±0.40** | 59.38±0.36 | 75.11±0.28 | 76.32±0.46 | – | – |
| GNNs | DGCNN (2019) | 84.0±6.7 | 58.3±7.0 | 72.9±3.5 | 76.6±4.3 | 69.2±3.0 | 45.6±3.4 |
|  | DiffPool (2018) | 79.8±7.1 | 60.8±7.0 | 73.7±3.5 | 75.0±3.5 | 68.4±3.3 | 45.6±3.4 |
|  | ECC (2017) | 75.4±6.2 | 55.7±3.3 | 72.3±3.4 | 72.69±4.1 | 67.7±2.8 | 43.5±3.1 |
|  | GIN (2019) | 84.7±6.7 | 58.8±5.5 | 73.3±4.0 | 75.3±2.9 | 71.2±3.9 | 48.5±3.3 |
|  | GraphSAGE (2017) | 83.6±9.6 | 60.1±4.7 | 73.0±4.5 | 72.9±2.0 | 68.8±4.5 | 47.6±3.5 |
|  | GraphiT (2021) | 82.2±6.3 | 58.1±10.5 | 75.6±4.9 | – | – | – |
|  | GWRNN (2023) | 83.4±5.6 | – | 74.9±3.5 | 75.6±4.6 | 72.8±4.2 | 49.0±2.9 |
|  | Co-GNN (2024) | – | – | 73.1±2.3 | – | 70.8±3.3 | 48.5±4.0 |
|  | GKNN-WL (2025) | 85.73±2.70 | 58.29±2.54 | 74.94±1.10 | – | 69.70±2.20 | 47.87±1.78 |
|  | GKNN-GL (2025) | 85.24±2.28 | 60.13±1.94 | 75.36±1.12 | – | 69.90±1.44 | 45.67±1.22 |
|  | DSMVPool (2026) | 81.39±1.98 | – | – | 80.91±2.14 | 72.89±2.98 | – |

*Table 3.* Per-epoch runtime and memory consumption comparison on benchmark datasets. OOM indicates out-of-memory.

|  | MUTAG | PTC_MR | PROTEINS | DD | IMDB-B | IMDB-M |
|---|---|---|---|---|---|---|
| Time (Transformer-Based) | 0.078s | 0.325s | 1.731s | OOM | 1.803s | 2.337s |
| Time (Unoptimized) | 0.032s | 0.049s | 0.178s | 0.893s | 0.178s | 0.274s |
| Time (Optimized) | 0.036s | 0.061s | 0.203s | 0.432s | 0.177s | 0.298s |
| Memory (Transformer-Based) | 352 MB | 404 MB | 6378 MB | OOM | 564 MB | 446 MB |
| Memory (Unoptimized) | 356 MB | 380 MB | 642 MB | 9716 MB | 424 MB | 382 MB |
| Memory (Optimized) | 338 MB | 340 MB | 372 MB | 476 MB | 364 MB | 342 MB |

## 4.3. The Runtime and Memory Consumption

To provide a more comprehensive evaluation of the proposed model, we further evaluate the computational efficiency and memory consumption of the proposed GI-GCN. Specifically, taking GraphiT (Mialon et al., 2021) as a representative Transformer-based baseline, we compare it with both the unoptimized and optimized implementations of GI-GCN in terms of per-epoch runtime and GPU memory usage.

We set the batch size to 64 for the DD dataset and 32 for all other datasets. The experimental results show that GI-

GCN achieves higher training efficiency while consuming less GPU memory than Transformer-based approaches. Notably, on the medium-scale PROTEINS dataset, GI-GCN demonstrates clear improvements in both training speed and memory efficiency compared with Transformer-based methods, highlighting its practical effectiveness. On the large-scale DD dataset, Transformer-based methods encounter out-of-memory (OOM) issues, whereas GI-GCN remains fully tractable, further demonstrating its scalability for large graph learning tasks. This efficiency gain mainly stems from two aspects. First, in terms of runtime, GI-GCN mod-

els global interactions through feature-correlation-driven quadratic optimization based on Dominant Set dynamics, without introducing additional learnable parameters. In contrast, graph Transformer models require numerous pairwise attention parameters, leading to higher update costs during training. Second, regarding memory consumption, the proposed feature-space reparameterization reduces the memory overhead to only $\mathcal{O}(F^2)$, which is independent of the number of nodes $N$, whereas Transformer-based methods typically incur $\mathcal{O}(N^2)$ memory complexity due to explicit global attention computation. This design enables GI-GCN to preserve effective global interaction modeling while substantially reducing computational and memory overhead.

In addition, we compare the unoptimized and optimized implementations of the GI-GCN. The unoptimized version explicitly constructs the global node-wise correlation matrix, whereas the optimized version performs an equivalent computation through feature-space reparameterization. Experimental results show that the optimized implementation introduces only a marginal runtime increase of approximately 1.27% per epoch, indicating negligible computational overhead. In contrast, the memory reduction is substantial, lowering GPU memory consumption to approximately 70.44% of that required by the unoptimized version. This advantage becomes particularly significant on large-scale datasets.

### 4.4. The Visualization of Node Importance

To intuitively illustrate the evolution of node importance learned by the proposed model, we visualize the node importance scores across different convolutional layers. We scale the node importance scores by the corresponding graph size so that the initial node importance is normalized to 1. In the visualization, the color intensity of each node indicates its relative importance, with darker colors corresponding to higher importance. As illustrated in Figure 2, the first two rows present example graphs from the PROTEINS dataset, while the remaining rows correspond to representative graphs from the IMDB-B dataset.

As the network depth increases, the node importance distribution gradually evolves from a relatively uniform pattern to a more concentrated one, where higher weights are consistently assigned to a small subset of representative nodes or substructures. This observation indicates that the Dominant Set dynamics steadily converge across layers, progressively reinforcing the influence of dominant nodes and guiding the model to focus on the most informative graph components. For molecular graphs in the Bio domain, the learned importance tends to focus on functionally relevant local substructures. In contrast, for the SN graphs, higher importance is often assigned to structurally prominent nodes or influential communities. This data-dependent and adaptive behavior further demonstrates the effectiveness and flexibil-

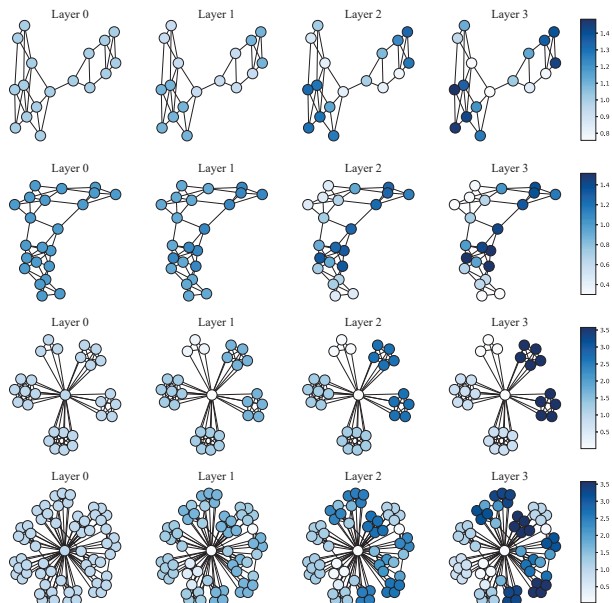

*Figure 2.* Examples of adaptive weight distribution.

ity of GI-GCN in modeling node importance.

Meanwhile, the visualization also confirms that GI-GCN preserves the original graph topology throughout the learning process. The proposed importance modulation only reweights node features and does not alter the adjacency structure or remove any edges. Consequently, the model enhances graph representation learning while maintaining complete structural consistency during message passing.

## 5. Conclusions

This paper proposes a novel GI-GCN for graph classification, which leverages the iterative dynamics of the Dominant Set to efficiently update the node importance distributions across the entire graph, guiding feature propagation and aggregation to progressively emphasize the dominant nodes during the convolution. This approach enables explicit modeling of arbitrary node relationships beyond local neighborhoods, while introducing no additional learnable attention parameters, which further enhances training efficiency. Moreover, through feature-space reparameterization, the proposed method significantly reduces the space complexity, enabling efficient global interaction modeling. Experimental results demonstrate the strong effectiveness of the proposed GI-GCN on the graph classification task. In future work, we plan to extend the proposed global interaction mechanism to more complex graph scenarios, such as heterogeneous graphs, dynamic graphs, and graphs with rich edge attributes. We will also explore more efficient iterative optimization strategies for global interaction modeling.

## Acknowledgements

This work is supported by National Natural Science Foundation of China (No. 62576371, 62576198, and 62536006), and the Open Project Foundation of Key Laboratory of Computation Intelligence and Chinese Information Processing of Ministry of Education and Key Laboratory of Data Intelligence and Cognitive Computing of Shanxi Province.

## Impact Statement

This paper presents work whose goal is to advance the field of Machine Learning. There are many potential societal consequences of our work, none of which we feel must be specifically highlighted here.

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

## A. The Theoretical Background on Dominant Sets

The Dominant Set is an optimization-based graph analysis framework that can be viewed as a continuous generalization of maximal cliques in weighted graphs (Pavan & Pelillo, 2007). It aims to identify subsets of nodes with high internal cohesiveness and strong separation from the remaining graph, thereby providing a principled formulation for discovering globally coherent structures.

Given a non-negative similarity matrix $\mathbf{A} \in \mathbb{R}^{n \times n}$, the Dominant Set problem is formulated as the following quadratic optimization over the probability simplex

$$\max_{\mathbf{x} \in \Delta} f(\mathbf{x}) = \mathbf{x}^\top \mathbf{A} \mathbf{x}, \tag{12}$$

where

$$\Delta = \left\{ \mathbf{x} \geq 0, \ \mathbf{1}^\top \mathbf{x} = 1 \right\} \tag{13}$$

denotes the standard simplex. Here, the vector $\mathbf{x}$ represents a soft membership assignment over all nodes, and each entry reflects the relative participation of a node in the dominant subset.

The Karush-Kuhn-Tucker (KKT) conditions imply the existence of multipliers $\lambda$ and $\mu_i \geq 0$, which can be expressed as

$$(\mathbf{A} \mathbf{x})_i - \lambda + \mu_i = 0, \quad \sum_i x_i \mu_i = 0. \tag{14}$$

Pavan and Pelillo (Pavan & Pelillo, 2007) established a one-to-one correspondence between Dominant Sets and strict local maximizers of this quadratic program in Equation (12) via weighted characteristic vectors. Specifically, a subset $S$ forms a Dominant Set if its corresponding vector $\mathbf{x}_S$ satisfies the KKT conditions and exhibits negative external coherence. This theoretical result provides a rigorous optimization foundation for interpreting the solution vector as a globally meaningful node-importance distribution.

In practice, the optimization can be efficiently solved using discrete-time replicator dynamics as

$$x_i^{(t+1)} = x_i^{(t)} \frac{(\mathbf{A} \mathbf{x}^{(t)})_i}{(\mathbf{x}^{(t)})^\top \mathbf{A} \mathbf{x}^{(t)}}. \tag{15}$$

This iterative update monotonically increases the objective value and converges to a stable point corresponding to a strict local optimum of the quadratic program, i.e., a Dominant Set. This provides a principled and optimization-grounded mechanism for identifying globally interacting nodes.

In the proposed GI-GCN model, we do not use the Dominant Set as a hard subset selector. Instead, the solution vector produced during iterative optimization is interpreted as a continuous global node-importance distribution. This distribution captures globally interacting and structurally important nodes, which is injected as a global prior into the subsequent graph convolution process to guide message passing. Since the entire process is driven solely by feature similarity and optimization dynamics, it introduces no additional learnable attention parameters, making it a parameter-efficient and interpretable mechanism for global interaction modeling.

## B. Ablation and Sensitivity Analysis

### B.1. Ablation Study

Table 4 reports the ablation analysis of GI-GCN components. We consider the following variants. **GI-GCN (Full)** denotes the complete model. **w/o Importance Modulation** removes the importance modulation term in Equation (7), so that node features are no longer reweighted by the learned global importance distribution before graph convolution. **Topology-Gated Correlation** replaces the global correlation matrix with topology-gated correlation by multiplying it with the adjacency matrix, which constrains global correlations to local topology and weakens the modeling of long-range interactions. **Random Initialization** initializes the iterative process with random node weights, whereas **Degree Prior Initialization** uses node degrees as the initialization prior. Finally, **Low-Dim (16)** and **High-Dim (64)** evaluate the effect of the feature dimension used in the equivalent reparameterization.

Among these variants, removing the importance modulation causes the most significant performance drop across datasets, confirming that the global feature modulation mechanism plays a central role in GI-GCN. The topology-gated correlation

*Table 4.* Ablation and sensitivity analysis of GI-GCN components.

| Setting | MUTAG | PTC_MR | PROTEINS | IMDB-B | IMDB-M |
|---|---|---|---|---|---|
| GI-GCN (Full) | **87.99** | **59.16** | **79.97** | **80.58** | **55.91** |
| w/o Importance Modulation | 81.54 | 57.29 | 75.16 | 72.01 | 51.08 |
| Topology-Gated Correlation | 83.11 | 57.38 | 70.41 | 73.74 | 49.40 |
| Random Initialization | 86.23 | 58.11 | 80.43 | 80.54 | 50.26 |
| Degree Prior Initialization | 83.89 | 57.99 | 79.89 | 79.33 | 47.22 |
| Low-Dim (16) | 85.15 | 58.45 | 79.16 | 80.39 | 54.85 |
| High-Dim (64) | 86.34 | 59.02 | 80.21 | 80.91 | 53.49 |

*Table 5.* Experimental results of feature perturbation.

| Perturbation Magnitude | 0 | 0.1 | 0.2 | 0.3 | 0.4 | 0.5 |
|---|---|---|---|---|---|---|
| PROTEINS | 0.805 | 0.794 | 0.788 | 0.785 | 0.776 | 0.753 |
| IMDB-B | 0.803 | 0.802 | 0.771 | 0.730 | 0.727 | 0.712 |

variant also consistently degrades performance, indicating that forcing global correlations to conform to local topology weakens the ability to capture long-range interactions. In addition, random or degree-based initialization leads to less stable performance across datasets, suggesting that the iterative optimization is sensitive to the initialization prior to some extent. Finally, experiments with different feature dimensions show that the model is relatively robust to this factor, and the adopted 32-dimensional setting provides a balanced trade-off between representation capacity and computational cost.

### B.2. Sensitivity Analysis on Node Importance

To further demonstrate that the learned node importance distribution captures graph-level discriminative signals, we conduct two sensitivity analyses. First, in the **feature perturbation experiment**, random noise is added to the original node features on two representative datasets, PROTEINS and IMDB-B, and the perturbation magnitude is gradually increased. Table 5 shows that stronger perturbations consistently reduce graph classification performance, confirming that node features contribute meaningfully to the learned graph representations.

Second, in the **importance masking experiment**, we use the node-importance distribution $\mathbf{p}$ produced by the iterative updates and remove either the top-$k$ high-importance nodes or the top-$k$ low-importance nodes. Table 6 shows that performance decreases as the removal ratio increases, and removing high-importance nodes causes a much larger degradation than removing low-importance nodes. This indicates that the learned importance distribution identifies nodes that are particularly discriminative for graph classification.

## C. Experiments on Large-Scale Graph Benchmarks

### C.1. Benchmark Description

To further evaluate the scalability of GI-GCN, we conduct experiments on three representative large-scale graph benchmarks, namely ZINC, OGB-MolHIV, and OGB-PPA. The ZINC dataset, proposed by (Dwivedi et al., 2023), is a benchmark for molecular property regression, containing approximately 12K graphs and evaluated by mean absolute error (MAE). The OGB-MolHIV and OGB-PPA datasets are from the Open Graph Benchmark (OGB) (Hu et al., 2020). Specifically, OGB-MolHIV is a molecular graph classification dataset with approximately 41K graphs, where the task is to predict whether a molecule inhibits HIV replication, evaluated by AUROC. OGB-PPA is a protein-protein association dataset containing approximately 158K graphs, mainly used to evaluate scalability under large graph workloads. The statistical information of these datasets is summarized in Table 7.

### C.2. Performance on Large-Scale Graph Benchmarks

To further evaluate the scalability and effectiveness of GI-GCN, we conduct experiments on two representative large-scale graph benchmarks, namely ZINC and OGB-MolHIV. We compare GI-GCN with a diverse set of representative graph learning models, including GCN (Kipf & Welling, 2017), GraphSAGE (Hamilton et al., 2017), GIN (Xu et al., 2019), GT (Dwivedi & Bresson, 2021), SAN (Kreuzer et al., 2021), GraphiT (Mialon et al., 2021), GraphGPS (Rampášek et al.,

*Table 6.* Experimental results of importance masking.

| Removal Ratio | 0 | 0.1 | 0.2 | 0.3 | 0.4 | 0.5 |
|---|---|---|---|---|---|---|
| PROTEINS Top-$k$ High | 0.805 | 0.738 | 0.489 | 0.245 | 0.227 | 0.272 |
| PROTEINS Top-$k$ Low | 0.805 | 0.796 | 0.794 | 0.782 | 0.776 | 0.753 |
| IMDB-B Top-$k$ High | 0.803 | 0.785 | 0.682 | 0.575 | 0.505 | 0.507 |
| IMDB-B Top-$k$ Low | 0.803 | 0.795 | 0.779 | 0.734 | 0.690 | 0.657 |

*Table 7.* Statistics of large-scale benchmark datasets.

| Dataset | #Graphs | Avg. Nodes | Avg. Edges |
|---|---|---|---|
| ZINC | 12,000 | 23.16 | 49.83 |
| OGB-MolHIV | 41,127 | 25.5 | 27.5 |
| OGB-PPA | 158,100 | 243.4 | 2,266.1 |

2022), MPNN+VN (Cai et al., 2023), GraphViT (He et al., 2023), Specformer (Bo et al., 2023), GRIT (Ma et al., 2023), and PST (Wang et al., 2024). The experimental results are reported in Table 8.

On the ZINC dataset, GI-GCN significantly outperforms classical GNN baselines and remains competitive with representative Transformer-based graph models such as GT and GraphiT, despite its lightweight optimization-driven design. Although its performance is still below some highly specialized graph Transformer architectures such as GraphGPS and GRIT, this gap is likely attributable to the fact that GI-GCN does not explicitly exploit the rich edge attribute information available in ZINC. In contrast, these specialized Transformer models typically incorporate sophisticated edge-aware mechanisms and involve substantially larger parameter budgets. By comparison, GI-GCN adopts a more lightweight optimization-driven global interaction framework with significantly fewer parameters, leading to favorable computational and memory efficiency. Under such lightweight modeling constraints, GI-GCN still achieves competitive performance against classical graph Transformer baselines, which highlights the effectiveness of the proposed feature-driven global interaction mechanism from the perspective of parameter efficiency. On OGB-MolHIV, GI-GCN outperforms several representative graph Transformer baselines, including GraphGPS and Specformer, demonstrating that the proposed feature-driven global interaction mechanism can achieve strong predictive performance even without learnable global attention modules. Overall, the experimental results demonstrate that GI-GCN achieves competitive performance on both large-scale benchmarks, indicating that the improvement of GI-GCN does not arise from overfitting to small-scale datasets but instead reflects the stronger representational capability brought by the proposed global interaction mechanism.

Compared with classical graph convolution models such as GCN and GIN, GI-GCN achieves substantial performance gains on both datasets, confirming the effectiveness of integrating Dominant Set dynamics into graph convolution. Moreover, compared with representative graph Transformer architectures such as GT and GraphiT, GI-GCN remains highly competitive despite its lightweight design. This result is particularly notable because GI-GCN is fundamentally different from Transformer-style models. Rather than introducing learnable pairwise attention, GI-GCN derives a continuous node-importance distribution through Dominant Set dynamics, which can be viewed as a quadratic-optimization-based continuous generalization of maximal cliques in weighted graphs. This optimization-driven process naturally captures global interactions without requiring additional attention parameters and injects global guidance signals into local message passing. Compared with other global modeling paradigms, GI-GCN also exhibits distinct advantages. For example, MPNN+VN introduces a virtual node for global communication, while GI-GCN directly learns node importance without introducing extra nodes. Specformer performs global spectral convolution, and PST adopts spatial global propagation, whereas GI-GCN provides a parameter-free optimization-based alternative. Furthermore, the proposed mechanism is not a form of linear attention, since the global interaction is derived from iterative optimization rather than attention approximation.

### C.3. Runtime and Memory Complexity Analysis

To further evaluate the computational efficiency of GI-GCN, we compare it with representative graph Transformer-based baselines GraphiT (Mialon et al., 2021) under the same experimental settings, including batch size, number of layers, and hidden dimension. The results are reported in Table 9.

As shown in Table 9, GI-GCN consistently reduces both runtime and memory consumption across all datasets. On average, GI-GCN achieves a memory reduction of 46.40% and a runtime reduction of 34.61%, demonstrating strong scalability on

*Table 8.* Performance comparison on large-scale graph benchmarks.

*(a)* Performance comparison on ZINC (MAE ↓).

| Model | Result |
|---|---|
| GCN | 0.367 |
| GraphSAGE | 0.398 |
| GIN | 0.526 |
| GT | 0.226 |
| SAN | 0.139 |
| GraphiT | 0.202 |
| GraphGPS | 0.070 |
| GRIT | **0.059** |
| GI-GCN (ours) | 0.211 |

*(b)* Performance comparison on OGB-MolHIV (AUROC ↑).

| Model | Result |
|---|---|
| GCN | 76.06 |
| GIN | 77.07 |
| SAN | 77.75 |
| GraphGPS | 78.80 |
| MPNN+VN | 76.87 |
| GraphViT | 77.92 |
| Specformer | 78.89 |
| PST | **80.32** |
| GI-GCN (ours) | 79.19 |

*Table 9.* Per-epoch runtime and memory comparison on large-scale datasets.

| Dataset | ZINC | OGB-MolHIV | OGB-PPA |
|---|---|---|---|
| Memory (GI-GCN) | 376MB | 378MB | 1828MB |
| Memory (Transformer-Based) | 446MB | 1874MB | 3246MB |
| Memory Reduction | 15.70% | 79.83% | 43.68% |
| Time (GI-GCN) | 1.04s | 3.61s | 30.47s |
| Time (Transformer-Based) | 1.55s | 4.78s | 56.89s |
| Time Reduction | 32.90% | 24.48% | 46.44% |

large-scale graph benchmarks.

This computational advantage mainly stems from two aspects. First, in terms of runtime, GI-GCN models global interactions through feature-correlation-driven Dominant Set optimization without introducing additional learnable pairwise attention parameters. In contrast, graph Transformer-based methods typically require expensive parameterized attention computations, resulting in higher update costs during training. Second, regarding memory consumption, the proposed equivalent reparameterization maps the Dominant Set update from the node space to the feature space, reducing the additional memory overhead from $O(n^2)$ to $O(f^2)$. As a result, the memory complexity becomes independent of the graph size. By comparison, graph Transformers usually incur $O(n^2)$ memory complexity due to global self-attention.

These results demonstrate that GI-GCN provides a favorable trade-off between predictive performance and computational efficiency. The efficiency gain highlights the advantage of the proposed optimization-driven global interaction mechanism for scalable graph representation learning.

