# OpenReview forum: "GI-GCN: Global Interacted Graph Convolutional Networks via Dominant Sets for Graph Classification"
_ICML.cc/2026/Conference — ICML 2026 regular_

### Official Review · Reviewer_GSPi · 2026-03-10

**Soundness:** 2
**Presentation:** 3
**Significance:** 2
**Originality:** 3
**Overall Recommendation:** 4
**Confidence:** 4

**Summary:**

This work addresses the limitations of Graph Convolutional Networks (GCNs) in capturing information beyond first-hop/local neighborhoods, which restricts the discriminative power of graph representations.
To overcome this, the authors integrate the Dominant Set, iteratively updated, to adaptively characterize the global importance distribution of nodes.
The proposed method emphasizes critical information at the graph level while enhancing the discriminative power of graph representations. Moreover, the framework’s spatial complexity is optimized, significantly reducing the memory overhead associated with modeling global interactions.
Empirical experiments on various graph benchmarks demonstrate the effectiveness of the proposed approach.

**Compliance With Llm Reviewing Policy:**

Affirmed.

**Final Justification:**

The responses addressed my concerns.

I select this score based on the trust to the authors that they will include results in rebuttal as well as the proposed discussion (partially included in the rebuttal) in the revised version.

**Key Questions For Authors:**

As listed in the weaknesses.

Please add discussion and comparison.

**Limitations:**

No limitation mentioned.

**Strengths And Weaknesses:**

## Strengths


1. The work is well motivated and addresses the critical limitations of Graph Convolutional Networks.
2. The proposed method demonstrates good balance between space complexity and the representational power.


## Weaknesses

1. The benchmarks in the experiments are relatively small in terms of number of graphs (at most around 1.5K). This might obscure the real improvement on representational power from the overfitting issues. Suggest to include the larger benchmarks from Dwivedi et al., (2022), Dwivedi et al., (2022b).




2. Considering the highly related motivation and goals of Graph Transformers. The literature review on Graph Transformers is a bit outdated.
Shall include the later works such as Zhang et al., (2023), Ma et al., (2023), He et al., (2023), Wang et al., (2024). There also lacks the comparison against graph Transformers in the experiments. (especially considering that graph Transformers can well handle the benchmarks in the experiments)

3. Lack discussion on and experimental comparisons to several highly related works
   - MPNN+Virtual Nodes (Cai et al., 2023); this is especially highly related to the Dominant set. Also need to think about the connection to linear-attention as discussed in this work.
   - global convolution in spectral routine (Bo et al., 2023)
   - global convolution in spatial routine (Ma et al., 2024)

4. No empirical evidence to demonstrate the scalability due to the efficiency of the proposed method (such as on large-scale-graph datasets)

> I believe the proposed method shall have advantages against the related works. However, the discussion and comparison against them are crucial.

- Dwivedi, V. P., Joshi, C. K., Laurent, T., Bengio, Y., & Bresson, X. (2022a). Benchmarking Graph Neural Networks. Journal of Machine Learning Research.
- Dwivedi, V. P., Rampášek, L., Galkin, M., Parviz, A., Wolf, G., Luu, A. T., & Beaini, D. (2022b). Long Range Graph Benchmark. Adv. Neural Inf. Process. Syst. Track Datasets Benchmarks. Thirty-sixth Conference on Neural Information Processing Systems Datasets and Benchmarks
- Zhang, B., Luo, S., Wang, L., & He, D. (2023). Rethinking the Expressive Power of GNNs via Graph Biconnectivity. Proc. Int. Conf. Learn. Represent.
- Ma, L., Lin, C., Lim, D., Romero-Soriano, A., K. Dokania, P., Coates, M., H.S. Torr, P., & Lim, S.-N. (2023). Graph Inductive Biases in Transformers without Message Passing. Proc. Int. Conf. Mach. Learn.
- He, X., Hooi, B., Laurent, T., Perold, A., Lecun, Y., & Bresson, X. (2023). A Generalization of ViT/MLP-Mixer to Graphs. Proc. Int. Conf. Mach. Learn.
- Wang, X., Li, P., & Zhang, M. (2024). Graph as point set. Proc. Int. Conf. Mach. Learn.
- Cai, C., Hy, T. S., Yu, R., & Wang, Y. (2023). On the Connection Between MPNN and Graph Transformer. Proc. Int. Conf. Mach. Learn.
- Bo, D., Shi, C., Wang, L., & Liao, R. (2023). Specformer: Spectral Graph Neural Networks Meet Transformers. Proc. Int. Conf. Learn. Represent.
- Ma, L., Pal, S., Zhang, Y., Zhou, J., Zhang, Y., & Coates, M. (2024). CKGConv: General Graph Convolution with Continuous Kernels. Proc. Int. Conf. Mach. Learn.

---

> ### Author Rebuttal · Authors · 2026-03-31
>
> Dear Reviewer GSPi,
>
> Thank you for your careful reading and valuable suggestions. Below we provide detailed responses to your comments. We hope these clarifications address your concerns and better highlight the contributions of our work. We appreciate any additional feedback.
>
> **Response to W1:**
>
> Thanks for the suggestion. We have included experiments on larger-scale datasets, including ZINC (\~12K graphs) and OGB-MolHIV (\~41K graphs). The results show that the performance gains of GI-GCN are not due to overfitting on small datasets. GI-GCN outperforms classical GNNs and remains competitive with graph Transformer-based methods, indicating that the improvement comes from stronger representational capability. We will include these additional experiments in the revised version.
>
> Table 1: Performance comparison on ZINC.
>
> |Model|MAE ↓|
> |-|-|
> |GI-GCN (ours)|0.211|
> |GIN|0.526|
> |GraphSAGE|0.398|
> |GCN|0.367|
> |GT|0.226	|
> |GraphiT|0.202|
> |SAN|0.139|
>
> Table 2: Performance comparison on OGB-MolHIV.
>
> |Model|AUROC ↑|
> |-|-|
> |GI-GCN (ours)|79.19|
> |GCN|76.06|
> |GIN|77.07|
> |SAN|77.75|
> |GraphGPS|78.80|
> |Specformer|78.89|
> |MPNN+VN|76.87|
> |PST| 80.32|
> |GraphViT|77.92|
>
> **Response to W2:**
>
> Thanks for the suggestion. In Sec. 2.1, we already cover representative graph Transformer models such as Graphormer, SAT, and GraphGPS. We will extend the related work section to include discussions on recent models, including Graphormer-GD, GRIT, GraphViT, and PST.
>
> More importantly, GI-GCN is not a Transformer variant but introduces a different paradigm for global modeling. It leverages Dominant Set dynamics, a quadratic-optimization-based method that can be viewed as a continuous generalization of maximal cliques in weighted graphs, to derive a continuous node-importance distribution. The Dominant Set provides a natural mechanism for global interaction without requiring any learnable parameters. This distribution is injected into local message passing, enabling a unified modeling of global feature interactions and local topological aggregation. Moreover, via reparameterization, GI-GCN reduces memory consumption from $O(n^2)$ to $O(f^2)$. This optimization-driven mechanism is fundamentally different from existing Transformer-style methods in both formulation and purpose.
>
> Empirically, we have compared GI-GCN with GraphiT on graph classification benchmarks, and further included comparisons with GT, GraphiT, GraphGPS, PST, GraphViT, and SAN on ZINC and OGB-MolHIV(see tables in Response to W1). The results show that GI-GCN remains competitive in comparison with graph Transformers. We will further extend these experiments to include the recently proposed Graph Transformer models.
>
> **Response to W3:**
>
> Thanks for the suggestion. Our GI-GCN leverages the Dominant Set algorithm to guide the graph convolution process. As we discussed in Response to W2, this optimization-driven mechanism is fundamentally different from existing research.
>
> Specifically, MPNN+VN introduces a virtual node for global communication, while GI-GCN directly learns node importance via Dominant Set dynamics without extra nodes or attention. Specformer performs spectral global convolution, and PST/CKGConv adopts spatial or kernel-based designs, whereas GI-GCN provides a parameter-free and memory-efficient alternative. Moreover, it is also not a form of linear attention, as the global interaction arises from an optimization process rather than attention approximation.
>
> Empirically, we have compared with Specformer, MPNN+VN, PST, and GraphViT on OGB-MolHIV(see table 2 in Response to W1), showing competitive performance with lower parameter and memory costs. We will further expand these analyses.
>
> **Response to W4:**
>
> Thanks for the suggestion. We have already provided empirical evidence of scalability in Table 3 by comparing runtime and GPU memory usage. Specifically, compared to a representative Transformer-based method, GI-GCN achieves **53.8%–90.2%** reductions in computation time and **4.0%–94.2%** savings in GPU memory across multiple benchmarks. Importantly, this advantage becomes more pronounced on datasets with large graphs. For example, on DD, where individual graphs contain thousands of nodes, Transformer-based methods encounter OOM issues, while GI-GCN remains fully trainable, demonstrating strong practical scalability.
>
> We further validate this property on a larger-scale dataset, OGB-MolHIV. Under the same setting, GI-GCN requires only **378 MB** GPU memory and **3.61s** per epoch, whereas Transformer-based methods require **1874 MB** and **4.78s**. This confirms that the efficiency advantage of GI-GCN persists as the dataset scale increases.
>
> This scalability is primarily attributed to the design based on the dominant set, which enables global interactions via an optimization-based approach without requiring additional parameters. On the other hand, our reparameterization design reduces memory consumption. We will further strengthen the discussion.

---

> > ### Author Rebuttal · Reviewer_GSPi · 2026-04-02
> >
> > I am generally satisfied with the authors’ responses. Since they indicate that additional discussion will be included in the revised version—although it is not available during the rebuttal stage—I am inclined to select (b).
> > However, I will update my score based on my trust that the authors will will carry out their proposed revisions..
> >
> > Additionally, for the ZINC benchmark,
> > (1) it would be important to include results from GraphGPS and GRIT (0.070 and 0.059, respectively),
> > (2) could you provide a discussion of the potential reasons for the observed performance gap.

---

> > > ### Author Response · Authors · 2026-04-05
> > >
> > > Thank the reviewer for the continued engagement and the insightful suggestion. Below, we give our detailed responses.
> > >
> > > Question: For the ZINC benchmark, it would be important to include results from GraphGPS and GRIT (0.070 and 0.059, respectively), along with a discussion of the potential reasons for the observed performance gap.
> > >
> > > Our Response: Thanks for the suggestion. We acknowledge that our current performance on the ZINC dataset is lower than that of GraphGPS and GRIT (0.070 and 0.059, respectively). We analyze this gap from both the experimental setup and the underlying model design as follows.
> > >
> > > First, from the experimental perspective, our current results are obtained under a relatively basic configuration due to the limited time in the rebuttal phase. Thus, we have neither performed the extensive hyperparameter tuning for GI-GCN nor incorporated the dedicated mechanisms to exploit the rich edge attributes in ZINC (e.g., edge-aware message passing). In fact, the classical previous works have shown that such components are crucial for this benchmark, suggesting that there still remains considerable room for further improvements with our proposed GI-GCN.
> > >
> > > Second, from the model design perspective, the gap also reflects a fundamental difference in terms of the paradigm. Specifically, our proposed GI-GCN is not a Graph Transformer-based architecture, but instead builds upon a conventional GCN framework. Its global dependencies are modeled via a feature-driven optimization mechanism (dominant set), which can capture the node importance without introducing additional learnable parameters.
> > >
> > > More concretely, in terms of the parameter scale, our GI-GCN mainly involves feature transformation matrices with complexity on the order of $O(F_{in} \times F_{out})$. In contrast, Graph Transformer models (e.g., GraphGPS, GRIT) rely on the multi-head self-attention and the parameter complexity is typically $O(H \times F^2)$, where $H$ is the number of attention heads and $F$ is the hidden dimension. This cost is further amplified by stacking multiple layers, leading to total parameter counts typically in the million scale or higher.
> > >
> > > As a result, our proposed GI-GCN operates with a significantly smaller parameter budget, offering superior computational and memory efficiency. Under this lightweight modeling constraint, our method still achieves competitive performance compared to classical Graph Transformer models such as GraphiT and GT. This highlights the effectiveness of the proposed feature-driven global interaction mechanism in terms of parameter efficiency. We will further strengthen this discussion in the final version and incorporate the results of GraphGPS and GRIT, following the reviewer's suggestion.
> > >
> > > Hope the above responses are helpful to address your concerns, and we appreciate the reviewer again for providing us this important opportunity to further explain.

---

### Official Review · Reviewer_F5fp · 2026-03-12

**Soundness:** 4
**Presentation:** 3
**Significance:** 3
**Originality:** 3
**Overall Recommendation:** 5
**Confidence:** 4

**Summary:**

This paper aims to capture the highly correlated information between nonadjacent nodes. To this end, the authors propose a novel Global Interacted Graph Convolutional Network (GI-GCN) based on the dominant set method. The dominant set can adaptively characterize the global importance distribution of different nodes, and the associated convolution operation of GI-GCN is defined associated with global interacted nodes identified by dominant set. The proposed GI-GCN can enhance the discriminative power of graph representations through the new convolution operation. The authors also optimize the framework of GI-GCN, so that they significantly reduce the spatial complexity and guarantee the feasibility. Experiments are performed on some benchmark datasets, the results show that GI-GCN has better graph classification performance than state-of-the-art methods.

**Compliance With Llm Reviewing Policy:**

Affirmed.

**Key Questions For Authors:**

I wonder whether node subset identified by dominant will influence the original graph structure? Because as I see, if the adjacent nodes are not in the subset, the probability of these node will be zero? Then, the edge between them can be ignored during the convolution operation. How to guarantee that it will not delete important edges? Whether this will influence the effectiveness?

**Limitations:**

Yes

**Strengths And Weaknesses:**

Strengths

- The node information propagation of GI-GIN is not restricted by the neighbor nodes, some nonadjacent but global interacted nodes can also directly propagate structural information with GI-GCN.

- Identifying the global interacted nodes through dominant set seems interesting. And the optimization procedure of dominant set can be integrated within the computational framework of GI-GCN. This provide a novel end-to-end architecture for graph structure learning.

- The proposed GI-GCN has better computational efficiency and spatial complexity than Transformer-based GNN that can also identify global interacted nodes though attention strategy. And the classification performance of GI-GCN is also better than Transformed-based GNN.

Weaknesses

- The reason of using dominant set is not quite clear, why the authors choose this algorithm to perform the identification of interacted nodes over the global graph structure. Some theoretical justification should be provided.

- It seems that the distribution computed through dominant set is similar as a mask strategy that can also limit the information propagation between adjacent nodes, especially for the adjacent nodes belonging to the same set. But the authors do not provide any discussion or justification about the relationship between dominant set and mask strategy.

---

> ### Author Rebuttal · Authors · 2026-03-31
>
> Dear Reviewer F5fp,
>
> Thank you for your careful reading and valuable suggestions. Below we provide detailed responses to your comments. We hope these clarifications address your concerns and better highlight the contributions of our work. We appreciate any additional feedback.
>
> **Response to W1:**
>
> Thanks for the suggestion. We adopt the Dominant Set algorithm because it provides a clear and rigorous optimization framework for identifying highly cohesive and externally dissimilar node subsets over the entire graph, which aligns well with our goal of modeling interactions among globally important nodes.
>
> Specifically, the Dominant Set can be formulated as a quadratic optimization problem defined over the standard simplex. Its solution can be obtained via replicator dynamics and is guaranteed to converge to a strict local maximizer. Pavan and Pelillo[1] have established a one-to-one correspondence between Dominant Sets and the strict local optima of this quadratic program. Therefore, it is not a heuristic scoring mechanism, but a theoretically grounded method.
>
> In GI-GCN, we leverage this property and use the Dominant Set as an optimization-driven global modeling mechanism. It produces a continuous node importance distribution over the entire graph, which serves as an effective global guidance signal for graph-level tasks. Meanwhile, this process does not introduce additional learnable attention parameters, leading to better stability and interpretability.
>
> Therefore, the Dominant Set in our framework goes beyond simple node selection, but to exploit its continuous optimization nature to explicitly model interactions among globally important nodes, and inject this information as a global prior into the subsequent convolution process. We will further clarify this motivation in the revised version.
>
> [1] Pavan and Pelillo, Dominant Sets and Pairwise Clustering, TPAMI, 2007.
>
> **Response to W2:**
>
> Thanks for the suggestion. We argue that the Dominant Set is fundamentally different from mask strategies, although both may appear to perform selective emphasis on nodes or information.
>
> Mask strategies are typically discrete and hard structural operations, such as directly removing nodes or edges, or restricting information propagation via binary masks.
>
> In contrast, the Dominant Set in GI-GCN produces a continuous probability distribution defined on the simplex, which is derived from an optimization process jointly driven by global structure and node features, and reflects the relative importance of nodes across the entire graph. Importantly, this node importance distribution does not directly modify the adjacency structure. Instead, it is used to globally modulate node features, after which standard message passing is still performed on the original graph.
>
> We will include a more explicit discussion of the differences between Dominant Set and mask strategies in the revised version to further clarify this point.
>
> **Response to W3:**
>
> Thanks for the suggestion. Our GI-GCN does not modify the original graph structure and does not explicitly remove any edges. During the convolution process, we only apply a reweighting to node features, while message passing is still performed based on the original adjacency matrix. Therefore, GI-GCN preserves the graph topology and ensures that all edges consistently participate in message passing, without any form of structural pruning or edge removal.
>
> In addition, our method focuses on the node importance distribution obtained during the iterative optimization of the Dominant Set, rather than performing a hard subgraph selection after convergence. The learned node weights form a continuous distribution rather than binary indicators, and are progressively concentrated on more representative nodes across layers. This behavior corresponds to a form of soft emphasis rather than hard selection, allowing the model to highlight highly cohesive and discriminative nodes while preserving the information flow over the original graph structure, thus achieving a balance between efficiency and information fidelity.
>
> As illustrated in Fig. 2, we present the evolution of node weights across different layers, where the model gradually emphasizes key nodes while maintaining overall structural stability. We will further clarify this mechanism and its implications in the revised version.

---

> > ### Author Rebuttal · Reviewer_F5fp · 2026-04-02
> >
> > The author's detailed response thoroughly addressed my concerns, and I am very satisfied with it.

---

> > > ### Author Response · Authors · 2026-04-07
> > >
> > > Dear Reviewer F5fp,
> > >
> > > Thank you for your positive evaluation and insightful questions.
> > >
> > > We are glad that our clarifications regarding the optimization-driven motivation of Dominant Sets, its distinction from mask-based strategies, and the preservation of the original graph structure have addressed your concerns. We will further refine these explanations in the revised manuscript to improve clarity.
> > >
> > > Best regards,
> > > The Authors of Submission 8757

---

### Official Review · Reviewer_3fQX · 2026-03-12

**Soundness:** 3
**Presentation:** 4
**Significance:** 3
**Originality:** 3
**Overall Recommendation:** 6
**Confidence:** 5

**Summary:**

This paper proposes a new global interacted graph convolutional network (GI-GCN) to overcome the problem of the limited representation power of GNNs caused by treating nodes equally important as each other. Specifically, for each convolution layer, the global importance distribution of various nodes is used to adaptively modulate the importance weights of different node features, which is followed by the local message passing strategies. This newly developed convolution operator can capture the highly nonadjacent nodes via the dominant set optimization technique. Therefore, it effectively overcomes the deficiency of existing GNNs and enhances the discriminative power of graph representations. Experimental results demonstrate the performance of the proposed method.

**Compliance With Llm Reviewing Policy:**

Affirmed.

**Final Justification:**

The authors did a good job with the responses, and I think all my previous concerns have been addressed. I will maintain my positive judgment and consider improving the score accordingly.

**Key Questions For Authors:**

Please see my above Weaknesses.

**Limitations:**

The authors should discuss any limitations in the conclusion part and point out some future research directions.

**Strengths And Weaknesses:**

**Strengths**:

- This paper is novel and interesting, and it is clearly written and well organized.
- The technical part is sound and robust, with clear illustrations of the main idea of the proposed method using figures, mathematical formulations, and theoretical analysis of the properties.
- The experimental results show the effectiveness and efficiency of the proposed method.
- I believe it is a good work that contributes to the community of machine learning, in particular, the GNN field.

**Weaknesses**:
- The dominant set algorithm is a significant component of the proposed method, apart from the descriptions in Section 2.2. Can you mathematically show the basic principles of this method?
- To facilitate a better understanding of the motivation of this paper, can you add an illustrative example in the introduction part to show the investigated problem, including the challenges arising in most related works?
- Are there any limitations of the proposed method? Can you add some discussions?

---

> ### Author Rebuttal · Authors · 2026-03-31
>
> Dear Reviewer 3fQX,
>
> Thank you for your careful reading and valuable suggestions. Below we provide detailed responses to your comments. We hope these clarifications address your concerns and better highlight the contributions of our work. We appreciate any additional feedback.
>
> **Response to W1:**
>
> Thanks for the suggestion. Here we provide a clearer mathematical explanation of the Dominant Set. The Dominant Set can be viewed as a continuous generalization of maximal cliques in weighted graphs. It aims to identify a subset of nodes with high internal cohesiveness and strong separation from the rest of the graph, serving as a principled formulation of clusters. From an optimization perspective, Dominant Sets can be obtained by solving the following quadratic program. Given the similarity matrix $A$, we maximize:
>
> $f(x)=x^\top A x,\quad x \in \Delta=\{x \ge 0,\ \mathbf{1}^\top x=1\}.$
>
> The Karush-Kuhn-Tucker (KKT) conditions imply the existence of multipliers $\lambda$ and $\mu_i \ge 0$ such that:
>
> $ (Ax)_i - \lambda + \mu_i = 0,\quad \sum_i x_i \mu_i = 0. $
>
> Pavan and Pelillo[1] have established a one-to-one correspondence between Dominant Sets and strict local maximizers of this quadratic program via weighted characteristic vectors. Specifically, a subset $S$ forms a Dominant Set if its corresponding vector $x_S$ satisfies the KKT conditions and exhibits negative external coherence.
>
> In practice, this problem can be efficiently solved using discrete-time replicator dynamics:
>
> $x_i(t+1)=x_i(t)\frac{(Ax(t))_i}{x(t)^\top A x(t)}. $
>
> This update monotonically increases the objective and converges to a stable point, which corresponds to a strict local optimum of the quadratic program, i.e., a Dominant Set. This provides a principled and optimization-grounded mechanism for identifying globally interacting nodes. In our GI-GCN, we leverage this optimization process to learn a global importance distribution over nodes. This distribution captures structurally important nodes and is injected as a global prior into subsequent graph convolution, guiding message passing. We will further refine the mathematical exposition and improve clarity in the revised version.
>
> [1] Pavan and Pelillo, Dominant Sets and Pairwise Clustering, TPAMI, 2007.
>
> **Response to W2:**
>
> Thanks for the suggestion. We will include a more illustrative example in the introduction to enhance the motivation. A representative scenario in graph classification is molecular graphs, where labels are often determined by a few discriminative substructures, such as functional groups, local rings, or combinations of distant but interacting atoms. These patterns exhibit both local structural characteristics and long-range dependencies.
>
> However, existing methods have inherent limitations. GCNs mainly rely on local neighborhood aggregation and struggle to capture interactions among distant key nodes. GATs introduce attention but remain fundamentally a local message-passing scheme with limited global modeling capacity. Graph Transformers can model global dependencies, but typically incur higher parameter and memory costs, and may not fully exploit the inductive bias of local graph structures in graph classification tasks.
>
> Motivated by this, our goal is to unify global interaction modeling with local structural aggregation. Specifically, GI-GCN learns a global importance distribution via Dominant Set, and injects it into local message passing as a global prior. This enables a collaborative modeling of global key node interactions and local topology-aware aggregation, which directly addresses the limitations of existing approaches.
>
> **Response to W3:**
>
> Thanks for the suggestion. We will supplement the discussion of potential limitations and future research directions in the conclusion. First, our method is primarily designed for graph classification tasks and is particularly effective when key substructures and globally discriminative nodes are present. For more complex heterogeneous or dynamic graphs, or scenarios requiring stronger structural inductive biases, further extensions are needed. Second, although our reparameterization significantly reduces memory overhead, constructing and iteratively optimizing global interactions on very large graphs may still incur additional computational costs. Exploring further sparsification, sampling strategies, or hierarchical modeling represents promising directions for future work.

---

> > ### Author Rebuttal · Reviewer_3fQX · 2026-04-02
> >
> > The authors did a good job with the responses, and I think all my previous concerns have been addressed. I will maintain my positive judgment and consider improving the score accordingly.

---

> > > ### Author Response · Authors · 2026-04-07
> > >
> > > Dear Reviewer 3fQX,
> > >
> > > Thank you for your generous acknowledgment of our response.
> > >
> > > We are glad that our clarifications on the mathematical formulation of Dominant Sets, the illustrative motivation example, and the discussion of limitations have addressed your concerns. We will incorporate these improvements into the revised manuscript for better clarity and completeness.
> > >
> > > Best regards,
> > > The Authors of Submission 8757

---

### Official Review · Reviewer_UbXi · 2026-03-14

**Soundness:** 2
**Presentation:** 3
**Significance:** 2
**Originality:** 2
**Overall Recommendation:** 3
**Confidence:** 4

**Summary:**

This work introduces a framework for graph classification to address the limitations of standard GCNs that aggregates 1-hop information and treat them equally. To capture global interactions without computation and memory burden of graph transformers, the authors propose to integrate the iterative dynamics of the Dominant Set algorithm into the convolution process. The model calculates a feature-induced node correlation matrix and uses replicator dynamics to derive a global node importance distribution. This distribution adaptively modulates node features before standard local message passing occurs. This reduces the overall complexity from a factor of the number of nodes to a factor of the input feature dimensions.

**Compliance With Llm Reviewing Policy:**

Affirmed.

**Key Questions For Authors:**

Please address the weaknesses listed above.

**Limitations:**

Yes.

**Strengths And Weaknesses:**

Strengths

While Dominant Sets have been used in graph clustering and pooling, integrating their replicator dynamics directly into the layer-wise convolution in new.

The experiments aptly visualize how the node importance pattern changes as layers increase, giving more insights into broader aspects like interpretability.

Weaknesses

The authors talk about Dominant Sets uniquely "emphasizes critical information at the graph level" while enhancing discriminability isn't really new. There are similar ideas in prior works. However is it actually superior to transformers in global information aggregation? There is no theoretical guarantee and so this feels like an overstated claim, primarily because the memory overload in graph transformers can be overcome with minibatching (works like https://arxiv.org/pdf/2403.16030 or https://arxiv.org/pdf/2412.04738v1)

The authors claim that their method is much more computationally efficient. However, the size of the datasets are pretty small compared to the SOTA dataset sizes in the OBG leaderboard. For example ogbg-molhiv or ogbg-ppa can be two datasets the authors may have considered. In fact ogbg-molhiv is also imbalanced and may help to show the performance of the model in imbalanced settings.

The initial node correlation matrix is built entirely on the Pearson correlation of node features, explicitly ignoring graph topology. I know that the authors state this as a strength to capture non-local interactions, but it relies on the strong assumption that feature similarity directly equates to semantic importance. An ablation study examining how this behaves on heterophilic graphs would make the analysis much more robust.

There are no ablation/sensitivity studies to better understand the empirical validity of the proposed method. Some ablations can be: Skipping Eq7, multiply S with adjacency matrix (Checks topology-agnostic claim), initialize p with random or degree, feature dimension vs performance.

---

> ### Author Rebuttal · Authors · 2026-03-31
>
> Dear Reviewer UbXi,
>
> Thank you for your careful reading and valuable suggestions. Below we provide detailed responses to your comments. We hope these clarifications address your concerns. We appreciate any additional feedback.
>
> **Response to W1:**
>
> Thanks for the suggestion. Compared to the transformer-based graph models, our GI-GCN introduces a lightweight and efficient global interaction mechanism, achieving competitive global aggregation while using significantly fewer parameters and less memory.
>
> As discussed in Sec. 3.6, its advantages lie in global interacted graph convolution and efficient scalability. Specifically, our GI-GCN leverages Dominant Set dynamics to learn a global importance distribution, which is injected into local message passing. This enables a unified modeling of global feature interaction and local topological aggregation, rather than relying on purely global modeling or introducing additional attention parameters. Moreover, this process is parameter-free, and via reparameterization, the storage complexity is reduced from $O(n^2)$ to $O(f^2)$, indicating that the performance gain comes from a more efficient design rather than increased model capacity.
>
> We will further expand the discussion of related work in the revision and more clearly position the advantages of GI-GCN in comparison to existing approaches.
>
> **Response to W2:**
>
> Thanks for the suggestion. As shown in Table 3 of the paper, we already provide clear evidence that GI-GCN consistently outperforms Transformer-based methods in both runtime and memory usage. Specifically, compared to a representative Transformer-based method, GI-GCN achieves **53.8%–90.2%** reductions in computation time and **4.0%–94.2%** savings in GPU memory across five benchmarks. Notably, on datasets where individual graphs contain thousands of nodes (e.g., DD), our model remains trainable, while Transformer-based models encounter OOM issues.
>
> To further validate scalability, we evaluate on OGB-MolHIV under the same setting. GI-GCN requires only **378 MB** GPU memory and **3.61s** per epoch, whereas Transformer-based methods require **1874 MB** and **4.78s**, respectively. This demonstrates that GI-GCN achieves substantially better efficiency on large-scale datasets as well. We further include results on OGB-MolHIV(**see Table 2 in response to reviewer GSPi W1**), where GI-GCN outperforms classical GNNs and remains competitive with graph Transformers despite lower resource usage.
>
> **Response to W3:**
>
> Thanks for the suggestion. We clarify that our model does not ignore graph topology. The feature-based correlation matrix serves only to learn a global importance prior, which modulates node features before graph convolution. During the convolution process, we explicitly incorporate structural information through the adjacency matrix. Consequently, node representations in subsequent layers already incorporate both global feature correlation and local structural information. Thus, GI-GCN jointly models global interaction and local topology rather than being topology-agnostic.
>
> Regarding heterophily, it is primarily defined for node classification, whereas graph classification focuses on graph-level representation and discriminative substructures. Therefore, heterophily is not a standard criterion for graph classification. Instead, our method emphasizes identifying key nodes that often determine the graph-level label (e.g., functional groups in molecules or communities in social graphs), which is supported by the visualization in Figure 2. We will further strengthen the discussion of this aspect in the revised version.
>
> **Response to W4:**
>
> Thanks for the suggestion. We conduct comprehensive ablation and sensitivity analyses (Table 1). Removing Eq. (7) (w/o Importance Modulation) causes the largest drop, confirming the importance of global feature modulation. Topology-Gated Correlation (multiplying S with the adjacency matrix) also degrades performance, as it forces global correlations back into local topology, thereby weakening the ability to capture long-range interactions. Random or degree-based initialization leads to less stable cross-dataset performance. Finally, experiments with different feature dimensions show that the model is relatively robust to this factor, and the adopted 32-dimensional setting provides a balanced trade-off between representation capacity and computational cost. We will further include and elaborate this analysis in the revised version.
>
> Table 1: Ablation and sensitivity analysis of GI-GCN components.
>
> |Setting|MUTAG|PTCMR|PROTEINS|IMDB-B|IMDB-M|
> |-|-|-|-|-|-|
> |GI-GCN (Full)|87.99|59.16|79.97|80.58|55.91|
> |w/o Importance Modulation|81.54|57.29|75.16|72.01|51.08|
> |Topology-Gated Correlation|83.11|57.38|70.41|73.74|49.40|
> |Random Initialization|86.23|58.11|80.43|80.54|50.26|
> |Degree Prior Initialization|83.89|57.99|79.89|79.33|47.22|
> |Low-Dim (16)|85.15|58.45|79.16|80.39|54.85|
> |High-Dim (64)|86.34|59.02|80.21|80.91|53.49|

---

> > ### Author Rebuttal · Reviewer_UbXi · 2026-04-02
> >
> > Thank you for the detailed response. However, some of my original concerns are still not addressed.
> >
> > 1) Regarding claim that dominant sets are better at aggregating information, there are no theoretical justification.
> > 2) The authors conducted experiments on OGB-MIHIV, however, OGB-PPA, which is the larger dataset is still missing.
> > 3) The visualization in figure 2 only provides qualitative viewpoint and does not negate the original concern.

---

> > > ### Author Response · Authors · 2026-04-07
> > >
> > > **Response to Q1:** Thank you for the insightful comment. We clarify that our use of Dominant Set does not replace message passing, but enhances it by introducing a principled global interaction prior. For theoretical analysis, please refer to our response to reviewer 3fQX W1. Dominant Set is not heuristic, but grounded in a well-established quadratic optimization framework, enabling global interaction without learnable parameters. In GI-GCN, we use it to compute a global node-importance distribution, which is injected into message passing as a prior. Thus, our method strengthens message passing with a theoretically grounded global signal rather than replacing it. We will further clarify this explanation.
> > >
> > > **Response to Q2:** Thank you for your valuable suggestion. We agree that OGB-PPA is a challenging, large-scale benchmark. In response, we have supplemented a per-epoch runtime and memory comparison between GI-GCN and graph-Transformer-based baselines under the same experimental settings (batch size, layers, and hidden dimension). The results are reported in Table A. Across OGB-PPA and two other large datasets, GI-GCN reduces memory by 46.40% and runtime by 34.61%, demonstrating strong computational efficiency.
> > >
> > > We attribute the computational efficiency of our method to two aspects. First, in terms of runtime, our method models global interactions via a feature-correlation-driven quadratic optimization (dominant set) without introducing additional learnable parameters, whereas Graph Transformers require numerous extra parameters for pairwise attention, leading to higher update costs. Second, regarding memory consumption, our reparameterization strategy reduces memory usage, introducing only $\mathcal{O}(F^2)$ overhead independent of the node number $N$, while Graph Transformers incur $\mathcal{O}(N^2)$ memory complexity due to global attention.
> > >
> > > Due to the large scale of OGB-PPA, limited rebuttal time and resources, we regret that we cannot complete full experiments on this dataset before 7th April. However, results on ZINC and OGB-MolHIV (Tables 1 and 2 in our response to Reviewer GSPi W1) show that our lightweight design outperforms classical GNNs and remains competitive with graph Transformer-based methods. This indicates that our model maintains high performance with strong efficiency, and we are confident it will achieve promising results on OGB-PPA. We promise that full results will be included in the final version.
> > >
> > > Table A: Per-epoch Runtime and Memory Comparison on Large-Scale Graph Datasets.
> > > |Dataset|ZINC (~12K graphs)|OGB-MolHIV (~41K graphs)|OGB-PPA (~158K graphs)|
> > > |-|-|-|-|
> > > |Memory (GI-GCN)|376MB|378MB|1828MB|
> > > |Memory (Transformer-based)|446MB|1874MB|3246MB|
> > > |Memory Reduction|15.70%|79.83%|43.68%|
> > > |Time (GI-GCN)|1.04s|3.61s|30.47s|
> > > |Time (Transformer-based)|1.55s|4.78s|56.89s|
> > > |Time Reduction|32.90%|24.48%|46.44%|
> > >
> > > **Response to Q3:** Thank you for the insightful comment. We clarify that in our model, feature correlation is not assumed to be equivalent to semantic importance. Instead, it serves as a global importance prior that supplements long-range global interaction information, which is difficult for local message-passing mechanisms to capture, rather than replacing topological structure information. We conducted experiments to validate this:
> > >
> > > 1.	Feature Perturbation Experiment: We add random noise perturbations to the original node features and test on two representative datasets, PROTEINS and IMDB-BINARY, to assess the impact of perturbation magnitude. As shown in Table B, increasing the noise level consistently reduces graph classification performance, confirming the importance of node features in the model.
> > >
> > > Table B: Experimental results of the feature perturbation.
> > > |Perturbation Magnitude|0|0.1|0.2|0.3|0.4|0.5|
> > > |-|-|-|-|-|-|-|
> > > |PROTEINS|0.805|0.794|0.788|0.785|0.776|0.753|
> > > |IMDB-BINARY|0.803|0.802|0.771|0.730|0.727|0.712|
> > >
> > > 2.	Importance Masking Experiment: Based on the node importance distribution p from iterative updates, we remove the top-k high/low importance nodes and analyze the impact on graph classification performance (Table C). As the removal ratio increases, performance gradually drops, with a more pronounced effect when removing high-importance nodes. This shows that our model can capture the key nodes critical for graph classification.
> > >
> > > Table C: Experimental results of the importance masking.
> > > |Removal Ratio|0|0.1|0.2|0.3|0.4|0.5|
> > > |-|-|-|-|-|-|-|
> > > |PROTEINS Topk_High|0.805|0.738|0.489|0.245|0.227|0.272|
> > > |PROTEINS Topk_Low|0.805|0.796|0.794|0.782|0.776|0.753|
> > > |IMDB-BINARY Topk_High|0.803|0.785|0.682|0.575|0.505|0.507|
> > > |IMDB-BINARY Topk_Low|0.803|0.795|0.779|0.734|0.690|0.657|
> > >
> > > Furthermore, as shown in Table 1 of the rebuttal, removing the importance modulation mechanism leads to the largest performance drop, which quantitatively validates the effectiveness of this mechanism. We will add these analyses in the revision.

---

### Decision · Program_Chairs · 2026-04-30

**Decision:**

Accept (regular)

**Comment:**

**Overall Assessment:**

This paper received three positive reviews and one negative review. Overall, the work is above the acceptance bar and presents a thoughtful and technically sound approach to incorporating global information into graph convolutional networks without incurring the heavy computational and memory costs typical of graph transformer models. While some claims are overstated and the empirical evaluation could be strengthened, the core idea is novel, well executed, and empirically effective.

**Summary of Contributions**

The paper introduces GI-GCN, a graph classification framework designed to overcome a key limitation of standard GCNs: their reliance on uniform, local (1-hop) aggregation that fails to capture global node interactions. Instead of adopting full graph transformers, which are often computationally expensive, the authors integrate the dominant sets algorithm into the convolutional process itself. Specifically, GI-GCN computes a feature-induced node correlation matrix, applies replicator dynamics from the Dominant Set algorithm to derive a global node importance distribution, and uses this distribution to adaptively reweight node features prior to standard local message passing. This design enables the model to encode global structural and feature interactions while reducing overall complexity from a dependence on the number of nodes to a dependence on input feature dimensionality. The framework thus offers a lightweight alternative to transformer-style global aggregation.

**Strengths**

> The paper is clearly written, logically structured, and easy to follow.

> While Dominant Sets have previously been used in graph clustering and pooling, incorporating their replicator dynamics directly into layer-wise graph convolutions is novel.

> The formulation and optimization of the method are technically sound and well justified.

> Experimental results demonstrate competitive performance and improved efficiency over standard GCN baselines.

> Visualizations showing how node importance distributions evolve across layers provide useful interpretability.

**Weaknesses**

> The paper emphasizes that dominant sets uniquely capture “critical information at the graph level,” but similar ideas have appeared in prior attention and importance-weighting mechanisms.

> While transformers are criticized for memory overhead, this limitation can be mitigated via minibatching and other recent techniques. As a result, the comparison sometimes feels overstated.

> The computational efficiency claims are supported only on relatively small datasets. Experiments on larger benchmarks, such as OGB-scale datasets, were missing from the original submission (though the authors have agreed to add them in the camera-ready version).

> The paper lacks ablation analyses. The authors have acknowledged this and agreed to include such experiments in the revised version.

**Overall Evaluation**

This work addresses an important limitation of message-passing GNNs by introducing a principled and efficient mechanism for global interaction modeling. While the proposed method may sacrifice some representational power compared to more expressive architectures such as full graph transformers or global spectral methods, it offers an appealing trade-off between efficiency, interpretability, and performance.
The concerns raised are valid but do not undermine the core contribution. Given the authors’ willingness to address these points during the rebuttal and revision phase, the paper is judged to be above the acceptance threshold.